# Inferring a complete genotype-phenotype map from a small number of measured phenotypes

**Zachary R. Sailer**[1,2], **Sarah H. Shafik**[3], **Robert L. Summers**[3], **Alex Joule**[3], **Alice Patterson-Robert**[3], **Rowena E. Martin**[3]\*, **Michael J. Harms**[1,2]\*

**1** Institute for Molecular Biology, University of Oregon, Eugene, OR, United States of America, **2** Department of Chemistry and Biochemistry, University of Oregon, Eugene, OR, United States of America, **3** Research School of Biology, Australian National University, Canberra, ACT, Australia

\* rowena.martin@anu.edu.au (REM); harms@uoregon.edu (MJH)

**Data Availability Statement:** All relevant data are within the manuscript and its Supporting Information files.

## Abstract

Understanding evolution requires detailed knowledge of genotype-phenotype maps; however, it can be a herculean task to measure every phenotype in a combinatorial map. We have developed a computational strategy to predict the missing phenotypes from an incomplete, combinatorial genotype-phenotype map. As a test case, we used an incomplete genotype-phenotype dataset previously generated for the malaria parasite's 'chloroquine resistance transporter' (PfCRT). Wild-type PfCRT (PfCRT$^{3D7}$) lacks significant chloroquine (CQ) transport activity, but the introduction of the eight mutations present in the 'Dd2' isoform of PfCRT (PfCRT$^{Dd2}$) enables the protein to transport CQ away from its site of antimalarial action. This gain of a transport function imparts CQ resistance to the parasite. A combinatorial map between PfCRT$^{3D7}$ and PfCRT$^{Dd2}$ consists of 256 genotypes, of which only 52 have had their CQ transport activities measured through expression in the *Xenopus laevis* oocyte. We trained a statistical model with these 52 measurements to infer the CQ transport activity for the remaining 204 combinatorial genotypes between PfCRT$^{3D7}$ and PfCRT$^{Dd2}$. Our best-performing model incorporated a binary classifier, a nonlinear scale, and additive effects for each mutation. The addition of specific pairwise- and high-order-epistatic coefficients decreased the predictive power of the model. We evaluated our predictions by experimentally measuring the CQ transport activities of 24 additional PfCRT genotypes. The $R^2$ value between our predicted and newly-measured phenotypes was 0.90. We then used the model to probe the accessibility of evolutionary trajectories through the map. Approximately 1% of the possible trajectories between PfCRT$^{3D7}$ and PfCRT$^{Dd2}$ are accessible; however, none of the trajectories entailed eight successive increases in CQ transport activity. These results demonstrate that phenotypes can be inferred with known uncertainty from a partial genotype-phenotype dataset. We also validated our approach against a collection of previously published genotype-phenotype maps. The model therefore appears general and should be applicable to a large number of genotype-phenotype maps.

**Funding:** This work was supported by a National Science Foundation CAREER Award (DEB-1844963 to MJH), an Australian Research Council Future Fellowship (FT160100226 to REM), and funding from the Australian National Health and Medical Research Council (Project Grant 1127338 and Fellowship 1053082 to REM; fellowship 1120690 to RLS). MJH is a Pew Scholar in the Biomedical Sciences, supported by The Pew Charitable Trusts. SHS was the recipient of an Australian Government Research Postgraduate Award. The funders had no role in study design, data collection and analysis, decision to publish, or preparation of the manuscript.

**Competing interests:** The authors have declared that no competing interests exist.

## Author summary

Biological macromolecules are built from chains of building blocks. The function of a macromolecule depends on the specific chemical properties of the building blocks that make it up. Macromolecules evolve through mutations that swap one building block for another. Understanding how biomolecules work and evolve therefore requires knowledge of the effects of mutations. The effects of mutations can be measured experimentally; however, because there are a vast number of possible combinations of mutations, it is often difficult to make enough measurements to understand biomolecular function and evolution. In this paper, we describe a simple method to predict the effects of mutations on biomolecules from a small number of measurements. This method works by appropriately averaging the effects of mutations seen in different contexts. We test the method by predicting the effects of mutations on a PfCRT—a macromolecule from the malarial parasite that confers drug resistance. We find that our method is fast and effective. Using a small number of measurements, we were able to gain insight into the evolutionary steps by which this macromolecule conferred drug resistance. To make this method accessible to other researchers, we have released it as an open-source software package: https://gpseer.readthedocs.io.

## Introduction

The genotype-phenotype map is an important tool for understanding evolution [1–13]. The distribution and connectivity of phenotypes in a map determines the accessibility of adaptive evolutionary trajectories [5, 7, 9, 14–16], tunes evolutionary dynamics [6,17–19], and alters population structure [20, 21]. However, characterizing genotype-phenotype maps can be very challenging because the size of the map expands exponentially as the number of mutations increases. For example, a map with four mutational sites, each existing in one of two states, includes 16 genotypes ($2^4$). By contrast, a map with 15 mutational sites consists of 32,768 genotypes ($2^{15}$). Given the time and cost of characterizing every genotype, researchers usually restrict their efforts to selected regions of the genotype-phenotype map [11, 22–24]. The ability to infer a complete genotype-phenotype map from a small dataset of experimentally determined phenotypes would therefore be extremely useful to a broad range of biology researchers.

In this study, we sought to infer phenotypes in a map of intermediate size, containing $2^8 = 256$ genotypes. This regime is particularly relevant given that the evolution of traits such as drug or pesticide resistance often involves 5–10 mutations (i.e. 32–1,024 genotypes) [4, 24–28]. A firm understanding of the evolution of these traits requires knowledge of the phenotypes of all (or most) of the genotypes. Complete combinatorial maps can reveal whether there were many or few accessible evolutionary trajectories between the wild-type and mutant isoform, whether the pathways were adaptive or required neutral steps, and why resistance sometimes evolves quickly [29–31], whilst taking decades in other cases [24, 32].

Exhaustive characterization of the phenotypes in a map containing hundreds of genotypes is often infeasible, particularly for phenotypes that are difficult to characterize by high-throughput methods. Yet such maps are also too small to be readily analyzed using sophisticated, data-hungry, machine-learning models that often require thousands or tens of thousands of observations. To address this shortfall, we have developed a straightforward approach to infer the missing phenotypes from an incomplete phenotype-genotype map. Our goal was to use combinatorial samples covering ≈20% of a map to infer the remaining phenotype

values, with well-characterized uncertainty in our predictions. Such knowledge would allow robust and statistically-informed analyses of evolutionary trajectories through an inferred genotype-phenotype map.

As a model dataset, we studied the map for the acquisition of chloroquine (CQ) transport activity by the malaria parasite's 'chloroquine resistance transporter' (PfCRT) [24, 33]. CQ is a diprotic weak-base that diffuses into the parasite's digestive vacuole (pH 5.0–5.5), wherein it becomes protonated and accumulates to high levels [34]. Here CQ exerts its antimalarial effect by preventing the detoxification of the heme generated from the parasite's digestion of host hemoglobin [35–37]. PfCRT is located at the membrane of the digestive vacuole [38]. Certain mutant isoforms of PfCRT confer CQ resistance by transporting CQ out of the vacuole and thus away from its antimalarial target [24, 33, 39–41]. The wild-type protein (PfCRT$^{3D7}$) lacks significant CQ transport activity, whereas the 'Dd2' isoform of PfCRT (PfCRT$^{Dd2}$) is the most commonly-studied of the PfCRT isoforms that confer CQ resistance [42]. PfCRT$^{3D7}$ and PfCRT$^{Dd2}$ differ at eight amino acid residues (Fig 1A). We can represent each genotype as a binary string, with PfCRT$^{3D7}$ being 00000000 and PfCRT$^{Dd2}$ being 11111111. Throughout the text we refer to the intermediate genotypes in this format.

The PfCRT genotype-phenotype map provided an excellent dataset for developing a predictive model. The phenotypes of 52 of the 256 possible combinations of the 8 mutations present in PfCRT$^{Dd2}$ had previously been characterized in the *Xenopus laevis* oocyte system by Summers et al. [24]. These genotypes were selected with the specific goal of identifying possible trajectories between PfCRT$^{3D7}$ and PfCRT$^{Dd2}$ [24]. These genotypes were scattered through the map, with two single-, seven double-, nine triple-, ten quadruple-, seven quintuple-, four sextuple-, and eight septuple-mutants. Each isoform was expressed at the surface of the oocyte and its capacity for CQ transport quantified using a radio-isotope uptake assay. These 52 isoforms constitute 20% of the PfCRT genotype-phenotype map (Fig 1A). Together, they revealed several evolutionary trajectories that may have been traversed to achieve PfCRT proteins with high capacities for CQ transport [24]. One of these trajectories is shown in Fig 1B. Interestingly, each of these trajectories included at least one step in which the mutation either did not significantly alter the protein's capacity for CQ transport or caused a slight decrease in activity. This apparent lack of adaptive trajectories may help explain why CQ resistance took some years to evolve in the field and why CQ resistance is yet to be generated from wild-type parasites (e.g., '3D7' parasites) following CQ pressure *in vitro*.

The Summers et al. study uncovered several trajectories that led to PfCRT acquiring CQ transport, but a complete genotype-phenotype map is required to determine whether there are other accessible trajectories. Although there are 8! = 40,320 possible forward trajectories through this map, the measured phenotypes allow us to assess the accessibility of just 428 of these trajectories. This leaves 39,892 trajectories—98.9%—for which one or more mutational steps are missing. However, measuring all of the remaining phenotypes would be both costly and labor-intensive.

We therefore sought to build a predictive model of the PfCRT genotype-phenotype map. This approach would allow us to understand how PfCRT evolved into a drug resistance transporter without having to experimentally characterize all 256 phenotypes. The model incorporates the additive effects of mutations, a nonlinear scale, and a logistic classifier. By characterizing the uncertainty in our predictions, we also know the uncertainty in our evolutionary inferences. Finally, we validated our final model against a collection of previously published genotype-phenotype maps. The approach we describe here appears to be applicable to many genotype-phenotype maps. We have released our implementation of the model as an open source Python software package (GPSEER; https://gpseer.readthedocs.io).

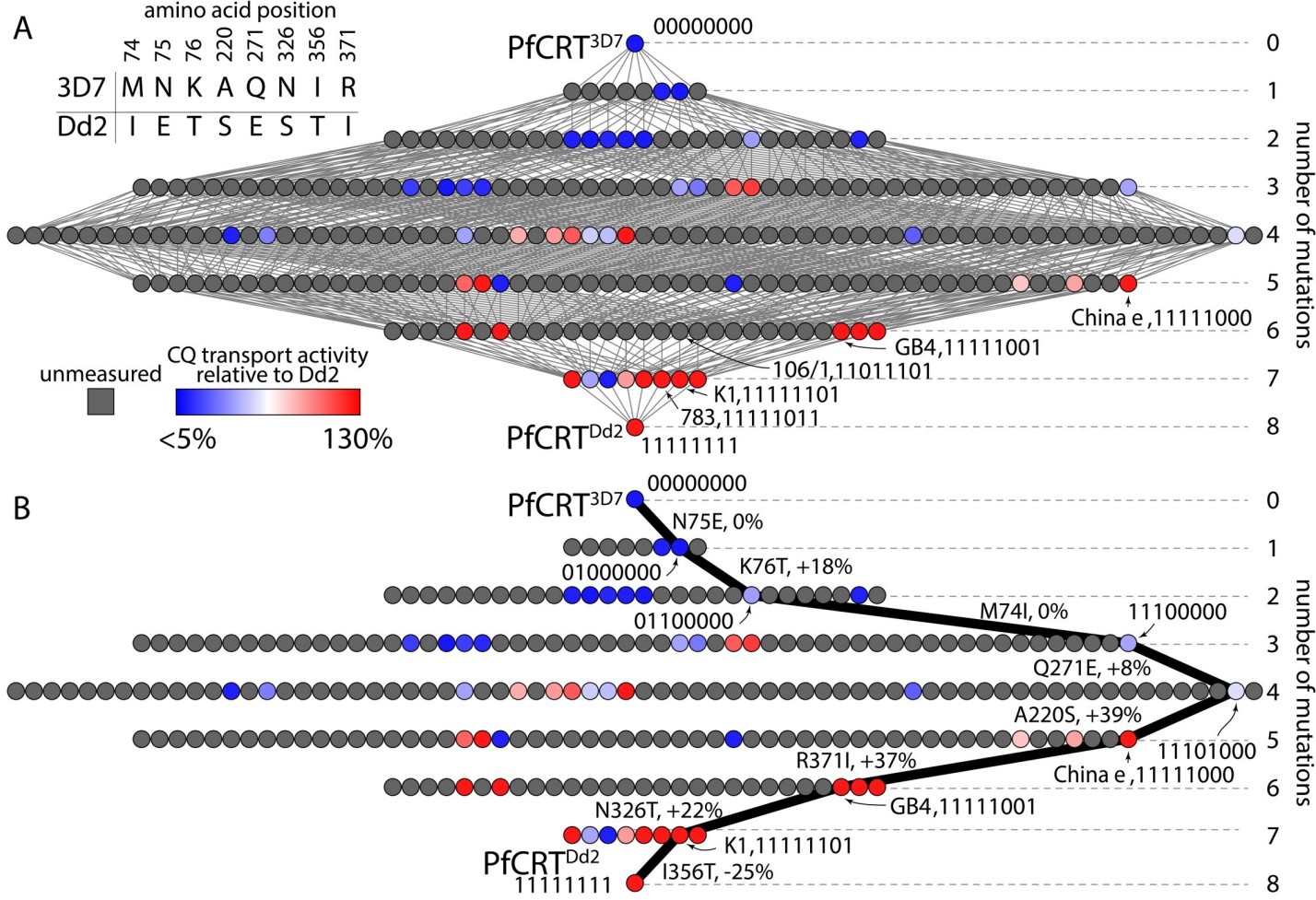

**Fig 1. Only 52 of the 256 possible genotypes between PfCRT³ᴰ⁷ and the CQ-transporting PfCRTᴰᵈ² isoform have measured phenotypes.** A) The table shows the amino acid residue differences between the wild-type (PfCRT³ᴰ⁷, 00000000) and mutant (PfCRTᴰᵈ², 11111111) isoforms of PfCRT. The network shows the complete set of genotypes between PfCRT³ᴰ⁷ and PfCRTᴰᵈ². Each node represents a different genotype with unique number and combination of the 8 amino acid residue differences between the two isoforms. Each edge connects genotypes that differ by a single mutation. Genotypes are in sorted combinatorial order, left to right, such that the second row contains 1000000, 0100000,. . ., 00000010, 00000001, the third row contains 11000000, 10100000, . . ., 00000101, 00000011, the fourth row contains 11100000, 11010000, . . ., 00001011, 00000111, etc. The gray nodes indicate PfCRT genotypes that have not had their CQ transport activities measured. The colors of the remaining nodes indicate experimentally determined CQ transport activities relative to the activity of PfCRTᴰᵈ² [24]. Values range from <5% (blue) to 130% (red). In addition to PfCRT³ᴰ⁷ and PfCRTᴰᵈ², the names and binary codes of five other field isoforms of PfCRT–"106/1", "GB4", "K1", "783", and "China e"–are indicated. B) One possible evolutionary trajectory from PfCRT³ᴰ⁷ to PfCRTᴰᵈ² that passes through only measured phenotypes. The mutations at each step are indicated next to the relevant edge, along with the effect on CQ transport activity. This trajectory passes through the PfCRTᶜʰⁱⁿᵃ ᵉ and PfCRTᴷ¹ isoforms. Five of the eight steps increase CQ transport activity, two have no effect, and the final step causes a decrease.

## Results

### Model development

We started with a linear, additive model that treats each PfCRT mutation as an independent perturbation to the quantitative phenotype of CQ transport activity. Where necessary, we added non-additive features to this model to better describe the experimental measurements.

We began by describing the phenotype of any given genotype as the sum of the effects of its mutations [43, 44]:

$$P_{obs} \sim P_{model}$$

where $P_{obs}$ is the observed phenotype and $P_{model}$ is a linear model with the form:

$$P_{model} = \beta_{ref} + \beta_1 x_1 + \beta_2 x_2 + \cdots$$

$\beta_{ref}$ is the phenotype of the reference genotype, $\beta_i$ represents the quantitative effect of mutation $i$, and $x_i$ is an index that encodes the presence or absence of a mutation in a given genotype (see Materials and Methods). We can estimate the effects of mutations in an additive model using linear regression:

$$P_{obs} = P_{model} + \varepsilon,$$

where $\varepsilon$ is the fit residual.

We fit this model to the 52 previously-measured phenotypes of PfCRT. This yielded an $R^2_{train}$ value of 0.65 between the model and the training dataset. To improve the model, we then examined the relationship between the observed phenotype ($P_{obs}$) and the predicted phenotype ($P_{model}$) (Fig 2A). This is an informative plot that can reveal mismatches between the linear, additive model and the statistical process that generated the data [7, 45, 46].

The first observation we made from Fig 2A was that the genotypes appeared to fall into two distinct classes. One class consisted of genotypes for which $P_{obs}$ remained close to zero, even as $P_{model}$ changed (red arrows, Fig 2A). The other class consisted of genotypes for which $P_{obs}$ increased monotonically with $P_{model}$ (blue arrow, Fig 2A). This suggested the presence of two distinct underlying statistical processes. The linear model fit poorly because two processes were being captured with a single model.

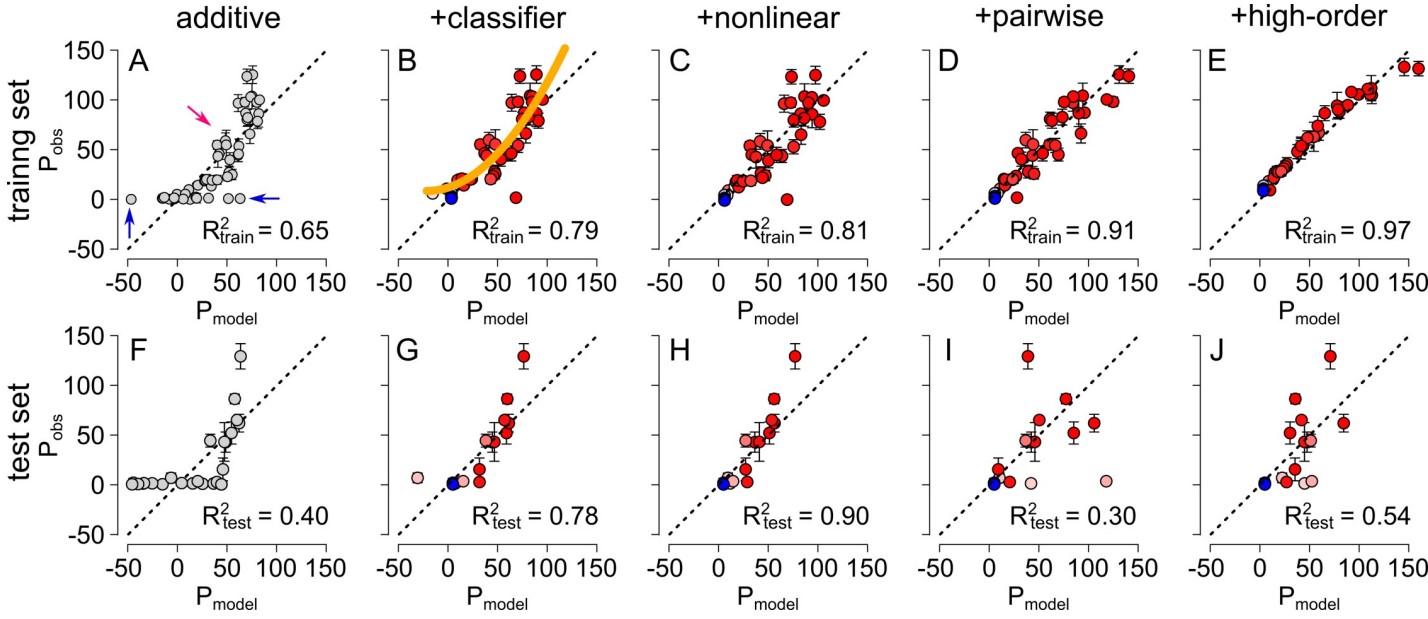

**Fig 2. Models can be trained on measured phenotypes to predict uncharacterized phenotypes.** The panels show the observed phenotype ($P_{obs}$) versus the predicted phenotype ($P_{model}$) for different input data and models. The vertical error bars represent experimental uncertainty (95% confidence interval on the mean) in the observed phenotypes. The resulting $R^2$ value is shown within each plot and the dashed line depicts a 1:1 relationship (i.e. perfect agreement). The top row (panels A-E) shows the quality of fit for the training dataset (the 52 published phenotypes [24]) and the bottom row (panels F-J) shows the quality of fit for the test dataset (24 newly-measured phenotypes). The models are arranged from left to right in order of increasing sophistication: additive (A,F), additive+classifier (B,G), additive+classifier +nonlinear (C,H), additive+classifier+nonlinear+pairwise epistasis (D,I), and additive+classifier+nonlinear+all epistatic orders (E,J). Symbol colors denote the probability a genotype belongs to either the *CQ-transporter* (red) or the *non-CQ-transporter* (blue) class. Gray symbols (panels A, F) show the additive model, prior to the application of a classifier. The blue arrows in panel A indicate data points for which the observed phenotype is zero and the predicted phenotype is nonzero. The red arrow indicates data points for which the observed phenotype is nonzero. The yellow line in panel B shows the spline fit to the data.

We addressed this issue by adding a classifier to the model. We defined the minimum threshold for the detection of CQ transport activity in the radio-isotope uptake assays as 5% of that measured for the PfCRT$^{Dd2}$ isoform (which served as the positive control in all of the experiments) [24]. We labeled genotypes with $P_{obs} \geq 5\%$ as *CQ-transporters* and $P_{obs} < 5\%$ as *non-CQ-transporters*. We then constructed a logistic classifier that predicted the class of any genotype given its sequence (see Materials and Methods). (Although we used a logistic classifier here, there are a number of ways such a classifier could be constructed; see Discussion). We trained this model on the published PfCRT dataset and colored the data points on the $P_{obs}$ versus $P_{model}$ curve according to the likelihood of the genotype belonging to either the *CQ-transporter* or the *non-CQ-transporter* classes (red and blue symbols in Fig 2B, respectively). Note that the spread of data points indicated by the blue arrows in Fig 2A collapsed to a set of overlapping blue points in Fig 2B. We could then train the linear model against members of the *CQ-transporter* class. The resulting fit produced a $R^2_{train}$ of 0.79 (Fig 2B).

After we applied the classifier, we noticed a second feature of the $P_{obs}$ versus $P_{model}$ plot: a nonlinear relationship between $P_{obs}$ and $P_{model}$ (yellow line in Fig 2B). Such curvature arises when mutations are not additive in their effects on the phenotype, but instead exert changes that combine in a nonlinear fashion [45, 46]. We therefore sought to improve the model by linearizing the data by $2^{nd}$-order spline interpolation [46]. The yellow line in Fig 2B is the spline fit to the data; we used this spline to linearize the data. Simply put, we transformed the data so the spline was straight (see Materials and Methods), resulting in the transformed data shown in Fig 2C. This step increased the $R^2_{train}$ to 0.81.

The additive model, classifier, and nonlinear correction left 19% of the quantitative variation unaccounted for (the scatter from the line in Fig 2C). We next sought to account for the remaining scatter by incorporating epistatic interactions between specific mutations [45, 47–50]. We added pairwise interactions to the model and then fit the model parameters using lasso regression [50, 51]. This method uses L1 regularization to penalize the addition of unneeded parameters, thereby minimizing the number of new parameters included. This approach added 24 parameters and increased the $R^2_{train}$ to 0.91 (Fig 2D).

It has been noted previously that under-specifying epistasis—that is, ignoring three-way, four-way, and higher-order interactions—can lead to biased estimates of the low-order terms of a model [52]. We therefore built a complete, high-order model that included all possible interactions between the eight mutations (from pairwise through to eight-way) [45, 47–49]. We again used lasso regression to discard parameters that did not contribute to the fit. The high-order model had 70 more parameters than the pairwise model and had an $R^2$ of 0.97 (Fig 2E).

## Testing the model with newly-measured phenotypes

To test the predictive power of the model, we selected 24 uncharacterized PfCRT genotypes and measured their CQ transport activities using the *Xenopus* oocyte system (Fig 3). To create this set, we randomly selected 18 genotypes from the center of the map that contained between three and six mutations. We completed the set of 24 by adding the six previously unmeasured single-mutation genotypes. We included these genotypes because the model predicted that none of these genotypes would possess CQ transport activity. We wished to test this prediction because it implies that the first evolutionary step from PfCRT$^{3D7}$ did not improve CQ transport activity. The 24 new test genotypes consisted of six single-, three triple-, five quadruple-, five quintuple-, and five sextuple-mutants. Eleven of the genotypes were predicted to lack CQ transport activity; the remaining 13 were expected to possess between 9% and 74% of the activity exhibited by PfCRT$^{Dd2}$.

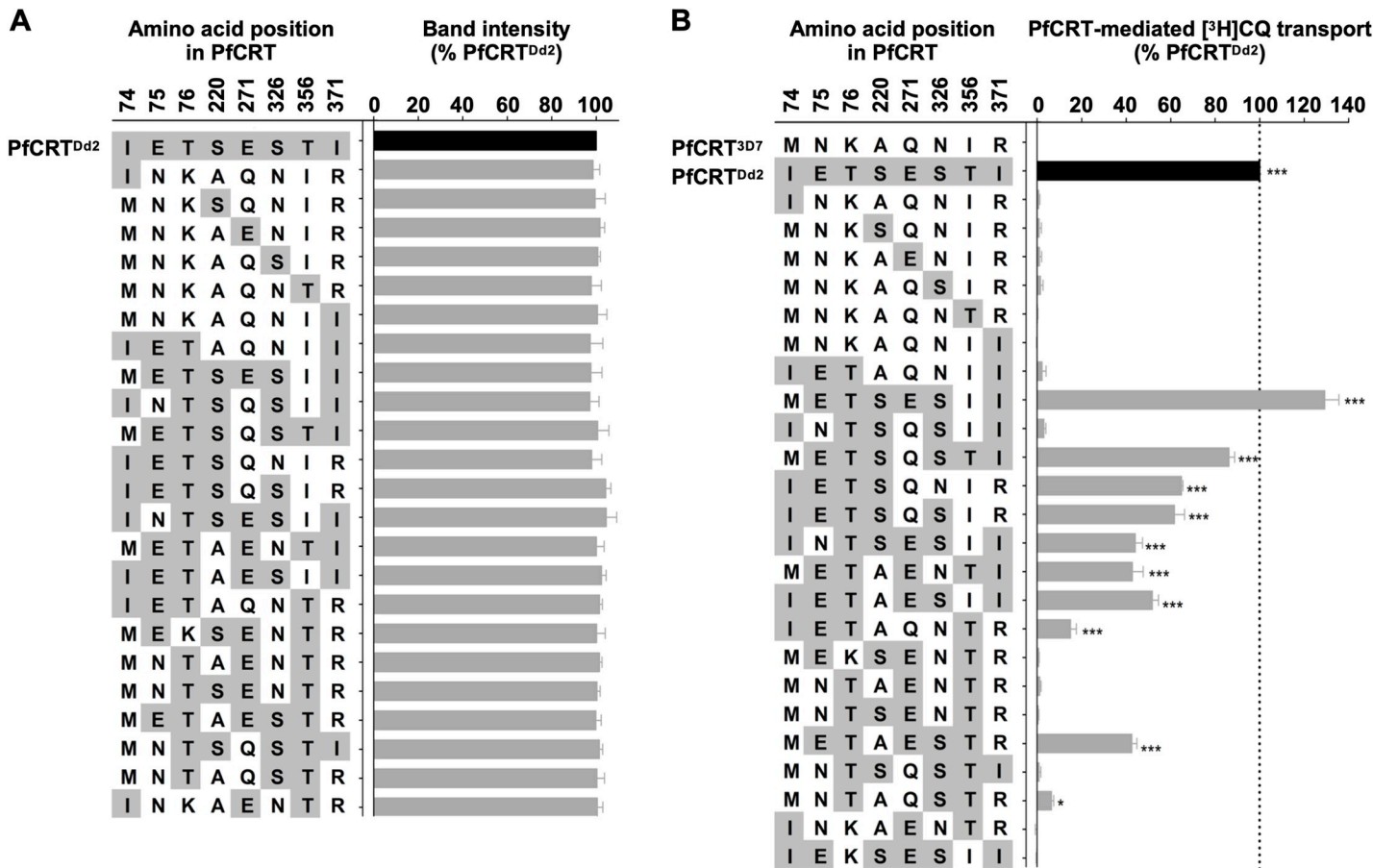

**Fig 3. Experimental characterization of the 24 combinatorial variants.** (A) The level of PfCRT protein in the oocyte membrane was semi-quantified using an established western blot approach [24]. The analysis included PfCRT[Dd2] as a positive control, to which the other band intensity values were normalized. Sample loading and transfer of the proteins were evaluated by total protein staining of the nitrocellulose membrane. The data are the mean + SEM of at least three independent experiments (performed using oocytes from different frogs), within which measurements were averaged from two independent replicates. There were no significant differences in expression levels between the 24 variants of PfCRT ($P > 0.05$; one-way ANOVA). That is, all of the PfCRT variants were present at similar levels in the oocyte membrane and any differences in CQ transport activity between these proteins are thus attributable to the mutations they carry, rather than differences in expression levels. (B) [³H]CQ transport was measured at pH 5.5 and in the presence of 15 μM unlabeled CQ. The direction of CQ transport in oocytes is equivalent to the direction of CQ transport in the malaria parasite (S1 Fig). The PfCRT-mediated component of CQ transport was calculated by subtracting the level of CQ accumulation detected in non-expressing oocytes (the negative control) from that measured in oocytes expressing a PfCRT variant. The rate of PfCRT-mediated transport was then expressed relative to that measured in oocytes expressing PfCRT[Dd2] (the positive control). The data are the mean + SEM of at least three independent experiments (performed using oocytes from different frogs), within which measurements were made from 10 oocytes per treatment. The asterisks denote a significant difference in CQ transport between oocytes expressing PfCRT[3D7] (the wild-type isoform) and oocytes expressing another variant of PfCRT: *, $P < 0.05$; ***, $P < 0.001$ (one-way ANOVA or Student's $t$-test). In both panels, the data for IEKSESII (i.e. the '106/1' isoform of PfCRT) was taken from Richards et al., 2016 [75]. The presence at the oocyte plasma membrane of the variants of PfCRT lacking significant CQ transport activity was confirmed via immunofluorescence assay (S2 Fig, S3 Fig).

Each genotype was expressed in the *Xenopus* oocyte system and the resulting rates of CQ transport were normalized to that of PfCRT[Dd2], where 100% indicates a capacity for CQ transport that is on par with this protein. The experimentally determined CQ transport activities of the 24 genotypes selected for testing ranged from 0% to 129 ± 6% (mean ± SEM; Fig 3). Fourteen of the genotypes failed to meet the minimum detection threshold of 5% CQ transport activity.

We then compared the predictions of the models trained on the 52 published PfCRT genotypes to the 24 new experimental measurements (Fig 2F–2J). The sequential addition of the classifier and nonlinear spline improved the fit to the test data, with the $R^2_{test}$ increasing from 0.40 to 0.78 and 0.90, respectively (Fig 2F–2H). However, the addition of pairwise epistasis

dramatically undermined the predictive power of the model. Although $R^2_{train}$ increased from 0.81 to 0.91 with the addition of pairwise epistasis (Fig 2D), $R^2_{test}$ fell from 0.90 to 0.30 (Fig 2I). The high-order epistatic model exhibited similar behavior (compare Fig 2E and 2J). This suggested that the epistatic interactions between mutations were not meaningful, but instead arose from over-fitting the model. These results are summarized in Fig 4A. Whilst $R^2_{train}$ increased monotonically as the model was made more complicated, $R^2_{test}$ increased with the addition of the classifier and nonlinear terms but then dropped precipitously upon the addition of epistasis.

To verify that this pattern was not due to unknown factor(s) specific to our training or test datasets, we repeated the analysis using k-fold cross validation. We combined all 76 genotypes (the original 52 plus the newly-measured 24) and sampled from this set to generate 5,000 pseudoreplicates of the 52-phenotype training dataset and 24-phenotype test dataset. The results replicated the pattern observed in the initial test and training datasets (Fig 4A), indicating that fitting epistatic interactions leads to poor predictive power (Fig 4B).

We next set out to identify the minimum number of measured phenotypes that would be required to fully train our best model (additive+classifier+nonlinear). We again employed a pseudoreplicate cross-validation approach, such that the 76 measured phenotypes were sampled to create training datasets that ranged from 10 to 66 genotypes. The model was then trained with each pseudoreplicate training set and evaluated for its abilities to reproduce the phenotypes of its training dataset and to predict the phenotypes of its matched test set. The results are shown in Fig 4C. $R^2_{train}$ decayed as the number of genotypes in the training set increased. This was because we began with almost as many parameters as observations and could therefore find parameter values that captured those few observations extremely well. As the number of observations increased, $R^2_{train}$ decreased because variation that was not accommodated by the model was encountered. By contrast, the predictive power of the model, which is indicated by $R^2_{test}$, steadily increased until it plateaued at $\approx$60 genotypes. This revealed that 60 or more measured phenotypes provided sufficient data to train the model.

## Model convergence and uncertainty

We next sought to understand why the model converged after $\approx$60 observations. In particular, we wished to understand if the convergence was a characteristic feature of the model or was instead a spurious result specific to the genotypes we sampled. To address this question, we

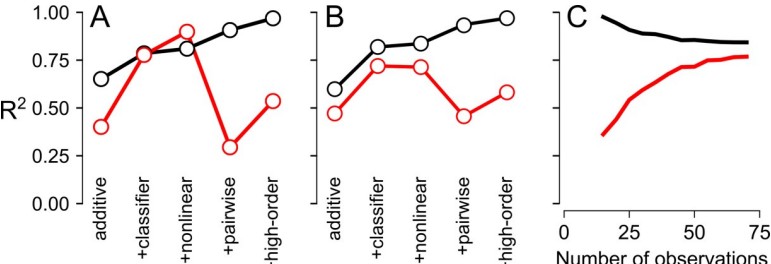

**Fig 4. The additive+classifier+nonlinear model captures the most variation without over-fitting.** In all panels, the lines denote $R^2_{train}$ (black) and $R^2_{test}$ (red). In panels A and B, the complexity of the model increases from left to right: additive model, plus classifier, plus a nonlinear spline, plus pairwise epistasis, and plus high-order epistasis. A) $R^2_{train}$ was calculated from the 52 phenotypes used to train the model and $R^2_{test}$ was calculated from the 24 newly-measured phenotypes. B) $R^2_{train}$ was calculated for pseudoreplicate training sets of 52 genotypes sampled from the 76 characterized genotypes, with $R^2_{test}$ calculated from the matched 24-genotype test datasets. C) The means of $R^2_{train}$ and $R^2_{test}$ converge as the number of observations in the pseudoreplicate training sets increase, reaching a plateau at ~60 genotypes (calculated with the additive+classifier+nonlinear model).

explored how additive models converged when applied to a generic genotype-phenotype map. Fig 5A illustrates how the additive model works conceptually. It averages the effect of each mutation over a large number of genetic backgrounds (Fig 5A). The difference in the effect of the mutation in different backgrounds is captured by the variance in its effect. The model converges when we have accurately determined the average effect of each mutation across all backgrounds. This view reveals why the predictive power begins to saturate as more observations are made. Once we have sufficient observations to resolve the average effect of each mutation across all backgrounds, the predictive power cannot improve.

We then used a set of simulations to ask how many phenotypes we would need to measure to construct a maximally predictive additive model. We constructed additive maps with different numbers of possible mutations at each site (ranging from 2 to 5 possibilities) and different numbers of sites (ranging from 6 to 8). We then injected normally distributed noise ranging in magnitude from 10% to 60% of the variation in the phenotype. This models epistasis with normally distributed residuals away from the additive description, matching the linearized experimental data. We simulated experiments where we measured one random genotype at a time, added it to our observations, and predicted the phenotypes of the remaining genotypes. We then plotted $R_{test}^2$ as a function of the average number of times we saw each individual mutation across all genetic backgrounds ($\langle n_{obs} \rangle$).

When plotted as a function of $\langle n_{obs} \rangle$, $R_{test}^2$ rapidly rises and then saturates at the magnitude of the epistasis in the map, independent of the number of mutations possible at each site and the number of mutated sites (Fig 5B). We next asked, as a function of the magnitude of the epistasis in the map, when our predictions would be within 0.05 of the best achievable $R_{test}^2$. This is indicated by the points on Fig 5B. We plotted these values as a function of the magnitude of the epistasis in the map. This analysis reveals a linear relationship between the average number of times we need to observe each mutation and the total epistasis in the map (Fig 5C). $R_{test}^2$ measures the amount of variation that is not accounted for by the model: in the context of a linearized and additive model, $R_{test}^2$ reports on the epistasis in the genotype-phenotype map.

This also provides a straightforward way to consider the uncertainty in each prediction. Because we linearized the data using a spline, the residuals between $P_{obs}$ and $P_{model}$ are normally distributed [7, 45, 46]. Each predicted phenotype therefore has the same, normally distributed uncertainty, which is captured by the standard deviation of the residuals between $P_{obs}$

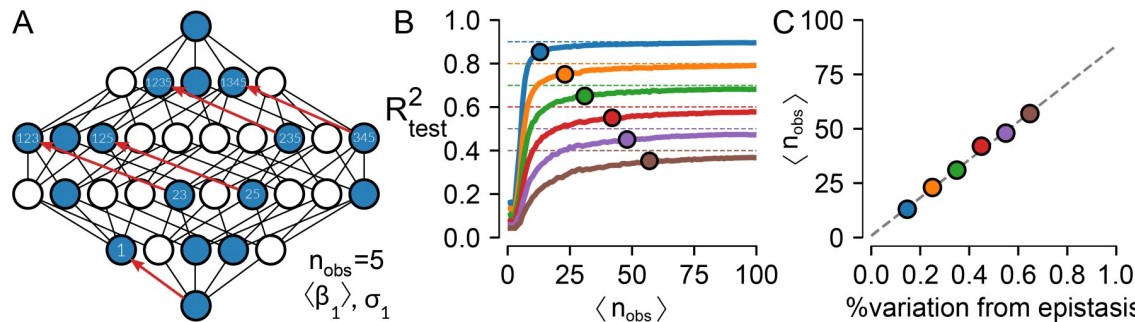

**Fig 5. Epistasis as uncertainty.** A) Schematic of a generic, partially characterized map. The nodes represent genotypes, some of which have been measured (blue) whereas others have not (white). Lines represent single point mutations. Given these observations, we measure the effect of "mutation 1" in five different backgrounds (red arrows) and can thus calculate the mean and variance in its effect across the map ($\langle \beta_1 \rangle$ and $\sigma_1$). B) $R_{test}^2$ versus the average number of times each mutation is seen in sampled genotype-phenotype maps with epistasis responsible for 10% (blue) to 60% (brown) of the variation in the maps. Points indicate where the $R_{test}^2$ is within 5% of the maximum predictive power of the additive model. C) A calibration curve indicating how many times, on average, one must observe each mutation in a map to resolve the additive coefficients with different fractions of epistasis.

and $P_{model}$. Even if a large amount of variation remains unexplained by the additive model, it is safely partitioned into a random normal distribution. This is the basic definition of epistasis given by Fisher [43], applied across the whole map. Thus, from the perspective of prediction, the magnitude of the epistasis is a measure of prediction uncertainty.

## PfCRT prediction uncertainty

Having established an approach to interpret the uncertainty of our model, we set out to quantify the uncertainty in our predictions for the PfCRT dataset. This model has two main sources of uncertainty: quantitative uncertainty in the predicted phenotypes from the additive model (as discussed above) and ambiguity in the classification of genotypes as *non-CQ-transporter* or *CQ-transporter*.

The error on our quantitative predictions is given by the fit residuals (Fig 5A, scatter from the 1:1 line in Fig 2C and 2H). This distribution is normal, has a mean of 0, and a standard deviation of 19%. The majority of our predictions will be closer to their true value than ±19%, but a few will be worse than this (for example, the outliers in Fig 2C and 2H). Across the whole map, we can treat this error by describing each phenotype as its predicted value plus a random number drawn from a normal distribution with a mean of 0 and a standard deviation of 19%. By repeating this sampling protocol multiple times, we can integrate over the uncertainty in our predictions.

We estimated the false-positive and false-negative rates for the classifier using a pseudoreplicate approach. We divided the 76 measured genotypes into 5,000 arbitrary training sets, each of which contained 61 genotypes paired with 15-genotype test sets. We then retrained the classifier on these pseudoreplicate sets and measured the false positive and false negative rates for the test sets. The average false positive and false negative rates were 6.2% and 5.2%, respectively.

## A small proportion of the genotypes in the PfCRT map possess CQ transport activity

We next used our model to better understand the distribution of phenotypes across the genotype-phenotype map separating PfCRT³D⁷ from its CQ-resistance-conferring counterpart (PfCRT^Dd2). We constructed a complete 256-genotype map by combining the 76 experimentally determined phenotypes with the remaining 180 predicted phenotypes. For our predictions, we used the additive+classifier+nonlinear model trained on the 76 experimentally determined phenotypes (Fig 6A). All predicted and measured phenotypes are listed in S1 File; the distributions of the phenotypes for the 76 experimental and 180 predicted genotypes are shown in S4 Fig.

We found that the majority of the genotype-phenotype map—170 of 256 genotypes—have no detectable CQ transport activity (Fig 6A). Genotypes with low transport activity are concentrated near PfCRT³D⁷, then become rarer as mutations accumulate (Fig 6B). We found that as the CQ-resistance-conferring mutations are introduced into PfCRT³D⁷, the average CQ transport activity increases from <5% (1 mutation) to 100% (8 mutations) (Fig 6B). Most importantly, none of the mutants containing a single mutation relative to PfCRT³D⁷ exhibit detectable activity, indicating that the first evolutionary step in this transition had very little effect on the capacity of PfCRT for CQ transport (Fig 6A and 6B). This finding, which was also a prediction of the original 52-genotype model [24], has been validated here by the characterization of the six previously-unmeasured single mutants (which all lack CQ transport activity; Fig 3).

One reason for the small number of genotypes capable of CQ transport activity was revealed by an analysis of the classifier. We counted the number of times each mutation appeared in

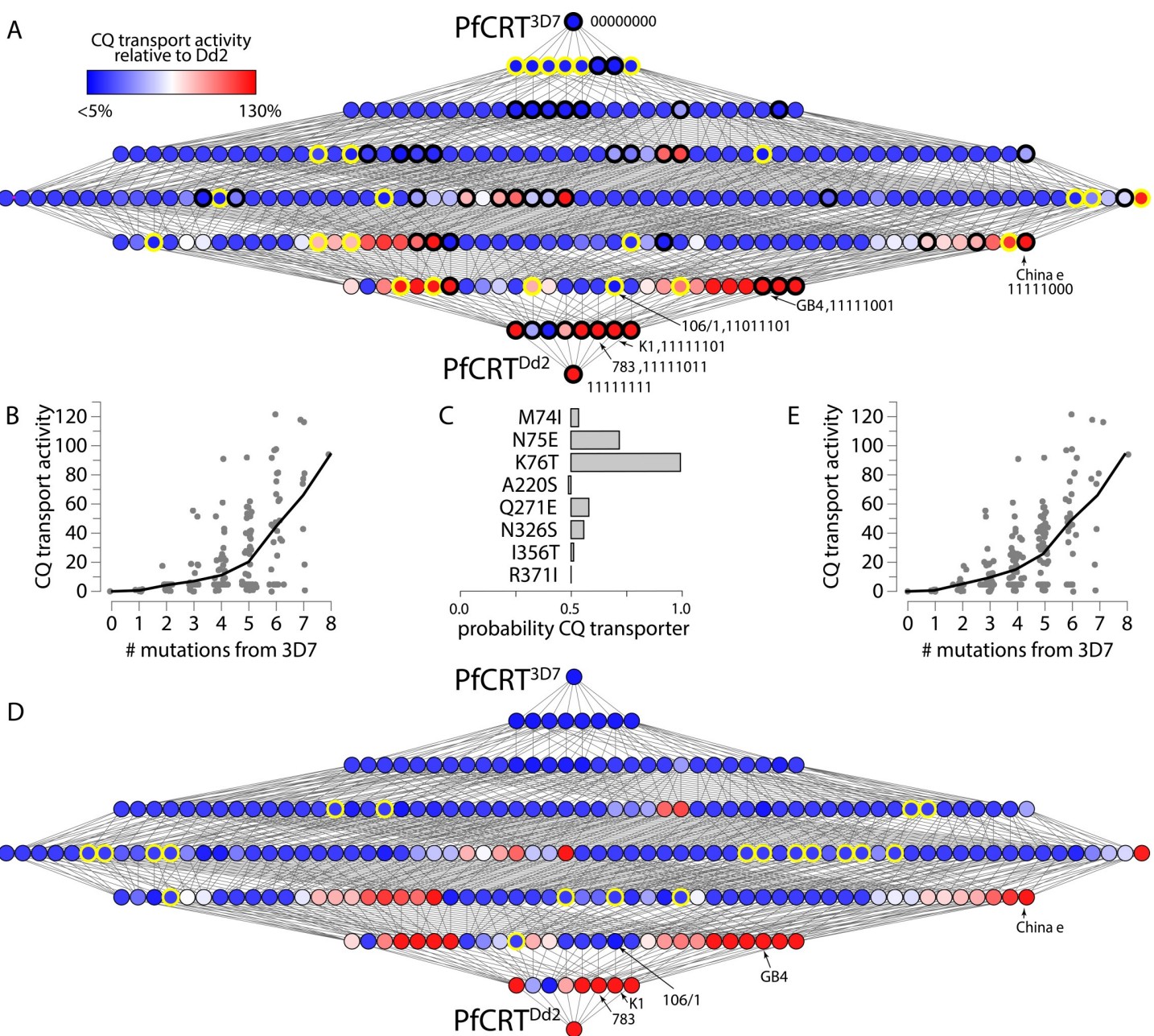

**Fig 6. CQ transport activities increases across the genotype-phenotype map.** A) The complete genotype-phenotype map with 76 measured phenotypes and 180 predicted phenotypes. As in Fig 1A, each node represents a genotype and each edge a mutation. The nodes outlined in black are the 52 genotypes that were measured by Summers et al., 2014 [24]; the genotypes measured in this study are outlined in yellow. B) Levels of CQ transport activity moving from PfCRT[3D7] (no mutations) to PfCRT[Dd2] (eight mutations). Each data point represents a genotype; the line is the mean across genotypes that have the indicated number of mutations. C) The contribution of each mutation to the classifier, where the x-axis is the probability that the mutation is found in genotypes that transport CQ. D) The complete genotype-phenotype map estimated using the simple K76T classifier. The nodes outlined in yellow indicate phenotypes that are predicted to have <5% CQ transport activity with this classifier, but not in the initial classifier from panel A. E) Predicted CQ transport activity as a function of the number of mutations calculated using the simple K76T classifier.

genotypes that the classifier placed in the *CQ-transporter* class. If a mutation did not contribute to the classifier, it would appear with equal frequencies in the *CQ-transporter* and *non-CQ-transporter* classes (probability = 0.5). We found that only the N75E and K76T mutations are

strongly enriched in *CQ-transporter*s (Fig 6C), consistent with their proposed critical roles in conferring CQ transport activity [24]. Only 64 genotypes contain both of these mutations, thus strongly constraining the map.

This analysis also revealed, however, that the classifier is too simplistic. It placed any genotype containing N75 in the *non-CQ-transporter* class and any genotype containing 75E in the *CQ-transporter* class. This is inconsistent with our experimental measurements, which contained several genotypes that exhibited detectable CQ transport without the presence of 75E (e.g. INTSESII, INTSESTI, INTAESIR, and MNTAQSTR). To test for the robustness of our conclusions to this over-aggressive classifier, we recalculated the map using a manual classifier that considered K76T but not N75E. For the manual classifier, any genotype with K76 was classified as a *non-CQ-transporter*, while any genotype with 76T was classified as a *CQ-transporter*. The map that was predicted with this classifier is shown in Fig 6D. This map has 20 more genotypes with detectable CQ transport activity relative to the original classifier (pink circles, Fig 6D). The two maps are, however, almost indistinguishable because the predicted activity of these new 'CQ-transporter' genotypes remains low (compare Fig 5C and 5E, which present CQ transport activities for the original classifier and the K76T classifier, respectively).

## PfCRT's evolution into a drug resistance transporter is highly constrained

We next used our estimated genotype-phenotype map to ask whether selection for CQ transport was sufficient to 1) allow the PfCRT$^{Dd2}$ isoform to evolve from the PfCRT$^{3D7}$ isoform whilst 2) populating intermediate PfCRT isoforms that have been observed in the field. These trajectories can be thought of as a null model, as the analysis does not account for other components of fitness (such as the requirement for the protein to maintain its natural function as it evolved into a bi-functional transporter) nor reversions along the evolutionary trajectory. A comparison of these predicted pathways with the fragments of PfCRT's mutational routes that can be inferred from the combinatorial genotypes–e.g., the PfCRT$^{China\ e}$ [11111000] and PfCRT$^{GB4}$ [11111001] isoforms–detected in field isolates will aid in identifying where mutations are likely to have become fixed for reasons other than the stepwise optimization of CQ transport.

To test the sufficiency of selection for CQ transport to produce PfCRT$^{Dd2}$, as well as the intermediate field isoforms, through the stepwise acquisition of mutations, we calculated the probability of all 40,320 possible forward trajectories connecting PfCRT$^{3d7}$ to PfCRT$^{Dd2}$. We employed the 'strong-selection/weak-mutation' model to calculate the trajectory probabilities [2, 16, 25, 27]. This model assumes that the parasite population was sufficiently large for selection to purge non-adaptive mutations (i.e. strong-selection) and that the mutations accumulated at a gradual rate, such that a mutant genotype became the dominant genotype before another mutant genotype arose in the population (i.e. weak mutation). We also assumed that each change arose as a single mutational event, and that there was a linear relationship between the CQ transport activity of a given PfCRT genotype and the fitness cost it is estimated to impart to the parasite (see Materials and Methods). This allowed us to calculate the relative fixation probability of a given mutation in each genetic background. We could then exhaustively sample all possible forward trajectories and record their relative probabilities.

The uncertainty inherent in each of the measured and predicted phenotypes will affect the calculated trajectories, but this can be accounted for using a simple sampling strategy to propagate uncertainty in CQ transport to uncertainty in evolutionary trajectories. To this end, we generated pseudoreplicate genotype-phenotype maps by drawing from the phenotype uncertainties (either experimental or estimated from the model). To account for uncertainty in the classifier, we flipped the classification of each genotype between the *CQ-transporter* and *non-*

*CQ-transporter* states according to the classifier's estimated false-positive and false-negative rates. To account for quantitative uncertainty in CQ transport activity, we randomly perturbed each phenotype by sampling from either the experimentally determined or predicted uncertainty for that phenotype. This resulted in each pseudoreplicate map having a different phenotype for each genotype. In some replicates, two genotypes may have very similar phenotypes; in others, the same two genotypes may have different phenotypes. This process integrates over our uncertainty about whether a given mutation is deleterious, neutral, or advantageous in a given background.

We calculated the probabilities of all possible trajectories between PfCRT$^{3D7}$ and PfCRT$^{Dd2}$ through each of the pseudoreplicate maps. This substantial set of sampled trajectories then allowed us to estimate the collection of adaptive trajectories that were consistent with both the experimentally determined and predicted values for CQ transport activity across the map.

We found that the evolutionary trajectories through the map were highly constrained, even given the uncertainty of our predicted phenotypes. The accessible trajectories are shown in Fig 7A. Of the 40,320 possible forward trajectories, 510 (1.3%) accounted for 99% of the total trajectory probability. We found that N75E and K76T were strongly favored to occur either first or second and that 90.4% of the total trajectory probability passed through the N75E, K76T-PfCRT$^{3D7}$ genotype. The remaining 9.6% of the of the trajectory probability was dispersed over a large collection of individual low-probability trajectories (Fig 7A).

We next considered whether this collection of trajectories between PfCRT$^{3D7}$ and PfCRT$^{Dd2}$ was compatible with genotypes observed in the field. We found that the probability of passing through the field isoforms was low (Fig 7A). Of the four key field genotypes, only PfCRT$^{783}$ [11111011] would be predicted to be visited with appreciable probability (15.2%) in trajectories that reach PfCRT$^{Dd2}$. The remainder had very low probabilities: PfCRT$^{China e}$ [11111000] (0.6%), PfCRT$^{K1}$ [11111101] (0.003%), and PfCRT$^{GB4}$ [11111001] (0.001%).

To understand why selection for CQ transport does not yield trajectories that populate these genotypes en route to PfCRT$^{Dd2}$, we undertook further analyses on the region of the map between N75E,K76T-PfCRT$^{3D7}$ and PfCRT$^{Dd2}$ (Fig 7B). This region contains 64 genotypes—the majority of which exhibit CQ transport activity—and includes the four field isoforms. There are many possible uphill trajectories through this region of the map (as noted above, it has 90.4% of the total trajectory probability). Of the 720 possible forward trajectories through this region, 142 account for 99% of the total trajectory probability across the whole map. The trajectories between N75E,K76T-PfCRT$^{3D7}$ and PfCRT$^{Dd2}$, and their probabilities, are listed in S2 File.

Three of the high-probability trajectories are shown in Fig 7B (rank-orders between 1 and 28). These trajectories pass through different regions of the map. For example, each trajectory begins by fixing a different one of the six possible steps after N75E,K76T (i.e. K74I, N326S, I356T, or R371I) (Fig 7C). These trajectories do, however, share common features. This becomes evident when the CQ transport activities of the genotypes are plotted in the order in which they arise within a given trajectory (Fig 7D). All three of the high-probability trajectories are the product of multiple small increases in the protein's capacity for CQ transport, thus achieving a relatively smooth and gradual transition to the CQ transport activity possessed by PfCRT$^{Dd2}$.

By contrast, the low-probability trajectories, such as those shown in Fig 7B (rank-orders between 131 and 526), often involve large increases in CQ transport activity followed by steps that do not improve—and sometimes lower—the protein's capacity for CQ transport (Fig 7E). This difference in the change in CQ transport activity between the high- and low-probability trajectories explains why the criteria applied to this model for the selection of CQ transport does not tend to yield trajectories that traverse the four field isoforms. For example, the

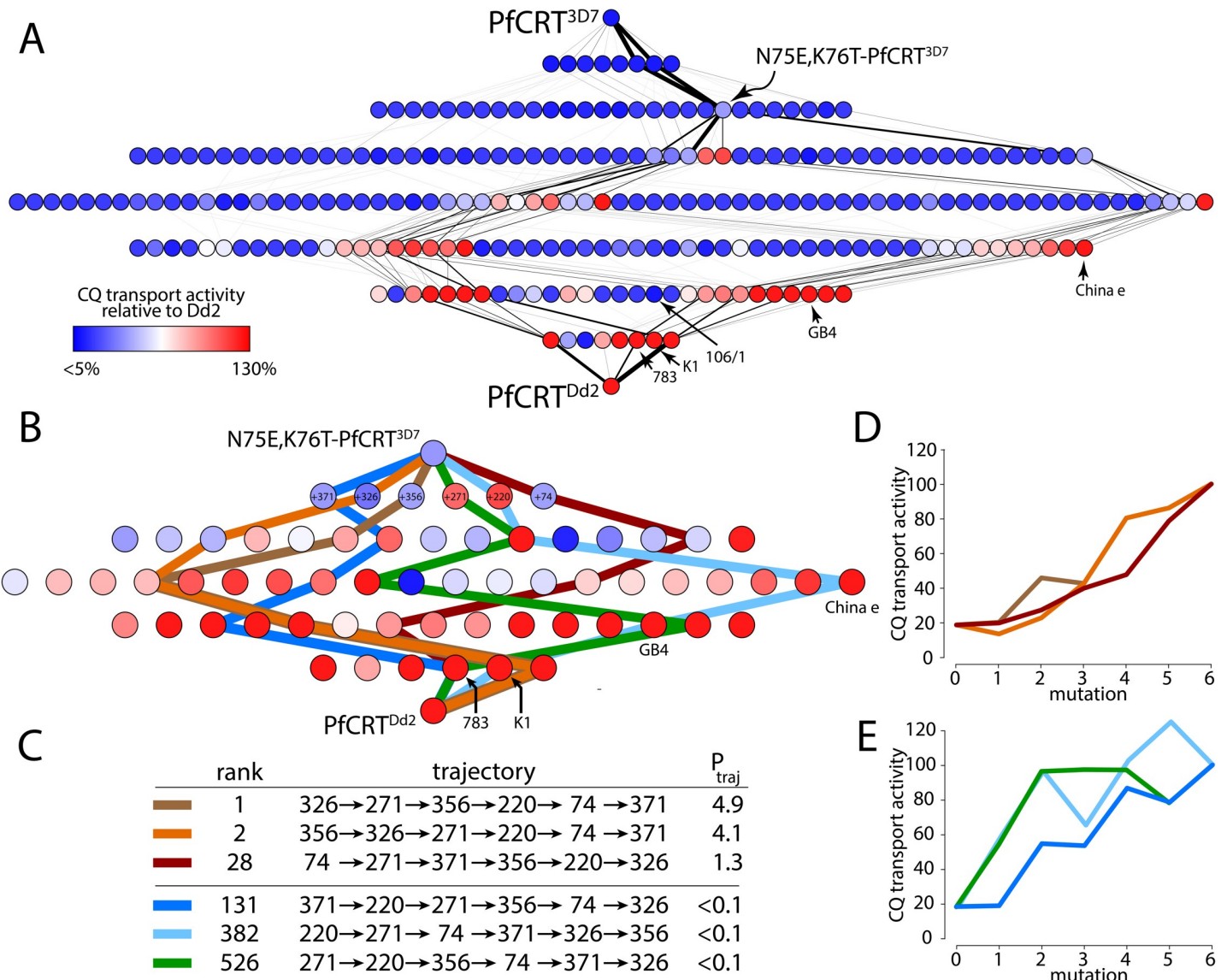

**Fig 7. Selection for increased CQ transport activity strongly constrains trajectories through the map.** A) Calculated trajectories through the complete genotype-phenotype map. The edges indicate the probability of a mutation over all possible trajectories from low (thin lines) to high (thick lines). The nodes indicate CQ transport activity as defined in Fig 1. B) Six example trajectories through the map between N75E,K76T-PfCRT[3D7] and PfCRT[Dd2]. C) The rank-order and mutational routes of the six trajectories shown in panel B. The $P_{traj}$ value indicates the relative probability of the trajectory. Panels D,E show CQ transport activity as a function of mutation step for the trajectories presented in panels B,C.

PfCRT[K1] [11111101] isoform is a local fitness peak within the map. If visited along a trajectory leading to PfCRT[Dd2] [11111111], CQ transport activity must decrease in the final step (mutation 5 to 6 in the light-blue trajectory, Fig 7E). Hence, if the PfCRT[Dd2] and PfCRT[783] [11111011] isoforms evolved from PfCRT[K1] [11111101] and PfCRT[GB4] [11111001], respectively, it is likely that a selection pressure other than CQ drove the addition of I356T to the latter proteins and the spread of resulting isoforms in field populations of the malaria parasite.

In this regard, it is worth noting that selecting purely for increased CQ transport activity—without assuming that the trajectory reaches PfCRT[Dd2]—would favor N75E,K76T-PfCRT[3D7] first acquiring A220S or Q271E. These two mutations impart the largest individual increases

in CQ transport (Fig 7E) and their early introduction into N75E,K76T-PfCRT[3D7] is likely to have conferred a significant selective advantage when levels of CQ pressure were high.

## Generality of the model

Having demonstrated the utility of our approach for the PfCRT genotype-phenotype map, we sought to determine its utility for predicting phenotypes in other genotype-phenotype maps. We therefore tested the models we investigated for the PfCRT dataset on 12 previously published, exhaustively measured, binary genotype-phenotype maps, ranging from 5–7 sites [4, 10, 14, 19, 28, 53, 54] (S2 Table). We then fit progressively more complex models to each dataset, as we did in Fig 4 for the PfCRT model. We observed similar behavior to all of these datasets: a nonlinear model with an additive scale gave the highest predictive power. Addition of epistatic coefficients led to reduced predictive power in every map we tested (S5 Fig). The parameters we used for the fits are reported in S3 File. This suggests that our approach—using a nonlinear model coupled to additive mutational effects—can be applied to other genotype-phenotype maps.

We also tested the utility of the model for making predictions in a much larger dataset: a partially sampled, experimental genotype-phenotype map characterizing the binding specificity of dCas9 to 23-base-pair oligonucleotides (Fig 8A) [55]. The published experiment sampled 59,394 of the $7 \times 10^{13}$ ($4^{23}$) possible oligonucleotides. Although all bases were sampled at all positions, there was bias towards a specific base at each position in the library (Fig 8A). As was performed for the PfCRT map, we linearized the dCas9 map with a 5th-order spline (Fig 8B and 8C). We then assessed the predictive power of the model by adding genotypes individually to a training set and evaluating our ability to predict the test set. We found that the model converged to $R^2_{test} = 0.62$ after about 4,000 measurements—that is, after we had seen each mutation at least 39 times. This finding is in good agreement with our simulated calibration curve (Fig 5D), which indicated that each mutation would need to be observed in approximately 40 different genetic backgrounds to saturate an additive model in which epistasis was responsible for 38% of the variation in the map (i.e. $R^2_{test} = 62\%$). The predictive power of this model is quite good considering its simplicity: we are able to predict phenotypes in the map to ± 38% given only 4,000 samples. This analysis was done in the context of training and test sets that were both sampled randomly from the same set of biased set of measured genotypes (Fig 8A). The predictive power of the model for a completely random, unbiased sample of genotypes—

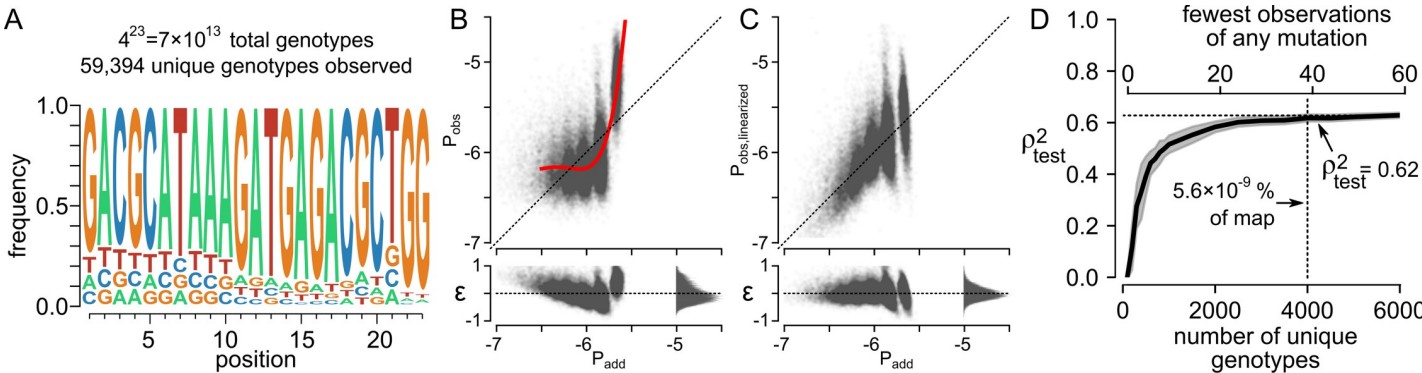

**Fig 8. A predictive, additive model can be trained on a large genotype-phenotype map.** A) Summary of the genotype-phenotype map [55].The map consists of 23 sites, each with four bases with the frequency at each site shown in the sequence logo. The total map has $7 \times 10^{13}$ genotypes; the phenotypes of 59,394 genotypes have been measured. B) Raw $P_{obs}$ vs. $P_{add}$ plot for the map. Each point is a genotype. The fit residuals are shown below the main plot. We fit an 5th-order spline to linearize the map (red curve). C) The linearized form of the map, with epistasis removed using the spline shown in panel B. D) A predictive model can be trained using approximately 4,000 genotypes. The bottom x-axis shows the number of unique genotypes used to train the model (sampled randomly); the top x-axis shows the fewest number of times any mutation was seen in that sample. $R^2_{test}$ was measured against the remaining 50,000+ genotypes not used to train the model.

and the general effect of bias in the training set on model power—remains an open question. Overall, however, this demonstrates that the model is general and useful for predicting phenotypes from sparsely sampled genotype-phenotype maps.

## Discussion

We have experimentally measured only 29.7% of the PfCRT$^{Dd2}$ [11111111] genotype-phenotype map, but have designed a model that can make robust conclusions about the distribution of CQ transport activity across the entire map and how this may have shaped the evolutionary process. The model we employ is straightforward, fast, and generally applicable to genotype-phenotype maps.

### Global features, not local epistasis, are useful for prediction

One key conclusion from our work is that estimating global features—a classifier and nonlinear scale—was much more powerful than estimating individual epistatic interactions between amino acid residues. This is in-line with previous observations about global features genotype-phenotype maps [45, 46]. Previous approaches to this problem have used a nonlinear function to describe global features [45, 46]; by adding a classifier on top of the nonlinear function, we better described the underlying distribution of function in the map and therefore gained predictive power. The classifier identified a step-function in the map, from non-functional to functional, mediated primarily by K76T. Once the genotypes were classified, the nonlinear function then identified the appropriate scale on which to sum mutational effects. Both the classifier and scale capture a large amount of the variation in a phenotype and are thus able to predict the unmeasured phenotypes with a relatively small number of parameters.

By contrast, modeling specific epistatic interactions between mutations drastically reduced predictive power (Fig 4A). The values of pairwise epistatic coefficients cannot be extracted reliably from sparse samples of maps that contain high-order epistasis because the high-order epistasis biases the low-order terms [52]. Since every map we have investigated exhibits high-order epistasis [45, 49, 56, 57], it follows that the addition of epistatic coefficients describing specific interactions will not provide reliable predictions. The PfCRT dataset demonstrates this concept quite clearly, as even the addition of pairwise epistatic coefficients strongly undermined our predictions (Fig 4A).

Another advantage to using global features, rather than specific epistasis, is that we can fit the model with relatively few measured phenotypes. We predicted that, given the magnitude of the epistasis in the map, observing each mutation across 20 combinatorial backgrounds was sufficient to estimate its additive effect (Fig 5C). The results reported here are in line with that finding; our predictions improved only marginally after observing 20% of the map (Fig 4C), which corresponds to each mutation occurring in at least 26 different backgrounds. This implies that the number of genotypes that need to be experimentally characterized to resolve the full map increases linearly with the number of mutations that are involved, even though the number of genotypes in the map scales exponentially.

Because the number of model terms increases linearly, our implementation should be effective even when applied to massive genotype-phenotype maps. The primary requirement is that the training dataset consists of a range of combinatorial genotypes, rather than consisting of genotypes that introduce different mutations individually into the same genetic background. This approach is necessary because it allows the background-dependent effects to be averaged and thus results in a much greater resolution of the additive coefficients.

A powerful aspect of using a global model, such as a spline, is that it normalizes the fit residuals. This approach gives all phenotype predictions the same, normally distributed uncertainty

(Fig 5A). These distributions can then be sampled to understand, in a statistical sense, what features of the map shape evolutionary trajectories independent of prediction uncertainty. This method sidesteps a key issue in completing partially measured genotype-phenotype maps: we will *never* have sufficient information to predict unmeasured phenotypes with perfect accuracy, regardless of sampling strategy [58].

## Future improvements

There are three main avenues by which the model could be improved. The first would be defining a more general framework for the classifier. For the PfCRT genotype-phenotype map, a logistic classifier readily identified the two groups of genotypes visually apparent in the data. In other maps, a different classifier may be much more effective. This could be a Gaussian process classifier [59, 60] (which we have, in fact, implemented in our software package), a variant of a principle-component analysis [61], or any number of other unsupervised classifier algorithms [62].

The second—and probably more important—way to improve the approach would be through a better description of the underlying nonlinearity that gives rise to the map itself. We fit the nonlinear scale with a spline [46]. This is a powerful approach to identify the curvature in data, but the spline is an empirical parameter that has no intrinsic, mechanistic meaning. Ultimately, it would be more desirable to define a model that describes the mechanism underlying the nonlinearity in the map [46, 63–65]. With such a model in hand, the global shape of the model could be described without resorting to the ad hoc approach of fitting a nonlinear scale. Furthermore, such a mechanistic model could potentially remove the apparent random epistatic interactions between certain mutations, thus yielding a highly predictive model.

Finally, a recent paper has identified a regression strategy that yielded predictive epistatic coefficients [66]. Such strategies, in combination with approaches such as classifiers and nonlinear splines, could allow the extraction of further information—and thus the development of better predictive models—from large genotype-phenotype maps.

## PfCRT evolution

Generating the complete genotype-phenotype map for the evolution of CQ transport activity between PfCRT[3D7] [00000000] and PfCRT[Dd2] [11111111] revealed that there are very few viable trajectories connecting the wild-type protein and the CQ-resistance-conferring isoforms. Several observations from the initial study [24] have been borne out and extended with this analysis. First, across the whole map, CQ activity requires K76T. This lends further support to the idea that this residue is directly involved in the transport of CQ. As has been noted previously, K76T entails the removal of a positive charge—which strongly suggests that this mutation contributes to the transport of the positively-charged CQ molecule via an electrostatic mechanism [24].

Secondly, because no single mutation results in detectable CQ transport, the first step in any trajectory beginning with the PfCRT[3D7] protein did not alter the CQ transport activity of PfCRT. This means that selection for CQ resistance could not drive the first mutation to a high frequency. This finding may help to explain why CQ resistance took a relatively long time to evolve in *P. falciparum* (compared with the emergence of *P. falciparum* resistance to other antimalarials and to the evolution of many different types of drug resistance in other organisms). One of several scenarios were likely required: 1) the first mutation arose within the population via drift or selection for another trait, followed by fixation of the second mutation through CQ pressure; 2) the second mutation occurred by chance in a small subset of the population that already possessed the first mutation; 3) the two mutations were brought together

by recombination across two alleles; 4) the same mutational event resulted in the introduction of both K76T and N75E.

Finally, there are very few viable trajectories through the map. Only 1% of the possible trajectories account for 99% of the total trajectory probability through the map (Fig 7A). This tight distribution of possible outcomes occurs despite the quantitative uncertainty in the CQ transport activity, suggesting that it is a robust feature of the map. This is an excellent demonstration of how adaptive evolution can be highly constrained whilst still reaching high-fitness peaks [4, 27, 59]. Additionally, these findings are consistent with other observations of relatively few trajectories being accessible for the evolution of drug resistance in the malaria parasite [16, 25].

Our results also show that selection for CQ transport alone is not sufficient to explain the evolutionary trajectories that appear to have been taken by PfCRT in field populations of the parasite. For example, the model generated high-probability pathways between PfCRT[3D7] and PfCRT[Dd2] [11111111] that do not pass through known field isoforms. This implies that other components of fitness—such as the ability of PfCRT to transport its natural substrate—must decrease the probability of some trajectories (by significantly reducing the overall fitness of one or more of the intermediates) and increase the favorability of others (by improving the overall fitness of intermediates that are neutral or even slightly deleterious with regard to CQ transport).

## Implementation of GPSEER

To facilitate phenotype predictions for other combinatorial genotype-phenotype maps, we have released our software as both a command line tool and application programming interface (https://gpseer.readthedocs.io). A user can provide a text file containing a list of genotypes and measured phenotypes: gpseer will automatically infer the phenotypes of uncharacterized genotypes, as well as provide uncertainties on those predictions. More advanced users can use the API to incorporate gpseer functions and methods into other analysis pipelines. The software package includes complete documentation, example data, and a tutorial to assist new users.

## Conclusion

These results show that a simple model can be effective in filling in unmeasured phenotypes in a genotype-phenotype map. Using methods that address global properties, such as classifying by genotype and fitting a nonlinear scale, allowed us to apply an additive model that explained 81% of the variation in an experimental genotype-phenotype map. By contrast, adding complexity via specific epistasis dramatically undermined the model's predictions. This implies that only a relatively small proportion of genotypes need to be experimentally characterized to infer the global properties of the genotype-phenotype map.

## Materials and methods

### Mathematical models

A linear epistasis model was used to decompose the PfCRT genotype-phenotype map into epistatic coefficients (up to $8^{th}$ order). We used the Hadamard model which uses the geometric center of the map as the coordinate origin [47–49]. Each genotype is made up of L sites. Each site has two possible states: "wild-type" or "derived" which are treated as a linear perturbation away from the origin of the map,

$$P = \beta_{origin} + \sum_{i}^{L} \beta_i x_i$$

where $\beta_{origin}$ is the origin of the genotype-phenotype map, $\beta_i$ is the effect of site $i$, and $x_i$ is 1 if site $i$ is "wild-type" and -1 if "derived".

We can add linear coefficients to describe interactions between mutations. For pairwise interactions, this has the form:

$$P = \beta_{origin} + \sum_i^L \beta_i x_i + \sum_{j<i}^L \beta_{ij} x_i x_j$$

where $\beta_{ij}$ is a pairwise epistatic coefficient. For the high-order model, the expansion continues:

$$P = \beta_{origin} + \sum_i^L \beta_i x_i + \sum_{j<i}^L \beta_{ij} x_i x_j + \sum_{k<j<i}^L \beta_{ijk} x_i x_j x_k$$

The model can be expanded all the way to 8$^{\text{th}}$-order interactions.

## Logistic classifier model

A logistic classifier was used to classify genotypes in the $P_{obs}$ vs. $P_{add}$ plot as *CQ-transporter* or *non-CQ-transporter*. A genotype was a *CQ-transporter* if its observed phenotype $P_{obs}$ was greater than a defined threshold, and a *non-CQ-transporter* if not. The threshold encodes the detection limit in our experiment.

We then trained a logistic model to classify the additive phenotypes given the observed phenotype classes. A genotype was classified as a *CQ-transporter* if it met the following condition:

$$\alpha_0 \beta_0 + \sum_i^L \alpha_i (\beta_i x_i) + \xi > 0$$

where $\beta_i$ are the additive coefficients, $\alpha_i$ are the odds that genotypes with mutation $i$ are inactive, and $\xi$ is the error in our model (which follows a logistic distribution) [67, 68].

## Nonlinear model

Global epistasis in the genotype-phenotype map was addressed by identifying a nonlinear function T that captures global curvature in the relationship between $P_{obs}$ and $P_{add}$,

$$P_{obs} = T(P_{obs}) + \varepsilon$$

where $\varepsilon$ are the fit residuals. We fit a 2$^{\text{nd}}$-order spline to the $P_{obs}$ vs. $P_{add}$ curve in the PfCRT map [46]. We then used this transform to linearize the map and extract linear epistatic coefficients.

## Trajectories

We calculated evolutionary trajectories as described previously [12], as implemented in the software package *gpmap* (https://github.com/harmslab/gpmap). Briefly, we calculated the probability of a given evolutionary trajectory as a series of independent, sequential fixation events. We assumed that the time to fixation for each mutation was much less than the time between mutations (the so-called strong selection/weak mutation regime) [2, 4]. We assigned fitness values to each genotype by assuming a linear relationship between CQ transport activity and selection coefficient. We used the formula: s = 0.1 x % CQ transport activity. We used the PfCRT$^{\text{3D7}}$ genotype as our reference to calculate relative fitness. Thus, relative fitness of PfCRT$^{\text{3D7}}$ (0% CQ transport activity) was 1.0, while the relative fitness of PfCRT$^{\text{Dd2}}$ (100% CQ transport activity) was 1.1. This selection coefficient is similar to values measured in direct

growth competition assays between parasites carrying either the PfCRT[3D7] or PfCRT[Dd2] isoforms [16]. We assumed a linear relationship between CQ transport activity and the selection coefficient.

## Software

All software is free and open source. The prediction models can be accessed and downloaded on Github (https://github.com/harmslab/gpseer). All of the code is written in Python and built on top of core scientific Python packages [69–72]. We have written documentation and examples of how to apply various types of experimental data (https://gpseer.readthedocs.io). The software takes a list of genotypes with their measured phenotypes and uses that to train the model. The code is written with a modular application programming interface, allowing users to try each layer of the model—classifier, nonlinear fit, and epistasis—with simple Python scripts (or integrated into a programming environment such as Jupyter). Because it is built on the existing epistasis package (https://github.com/harmslab/epistasis) [45], it can also handle arbitrary genotype alphabets, as the software builds the alphabet from the input data. Finally, all statistical tests and cross-validation are implemented within the software, allowing for any researcher to fit and select between different predictive models of the genotype-phenotype map.

## Experimental measurements

### Generation of PfCRT coding sequences and synthesis of cRNA

The coding sequences of each of the PfCRT isoforms were generated via site-directed mutagenesis using the primer pairs listed in S1 Table [24] and an approach described previously [24]. The mutations were introduced into PfCRT sequences generated in Summers et al., 2014. These sequences are codon-harmonized, free of endosomal-lysosomal trafficking motifs (to enable expression at the plasma membrane of *Xenopus laevis* oocytes), and are expressed in oocytes using the pGEM-He-Juel vector [73]. All of the resulting PfCRT coding sequences were verified by sequencing (undertaken by the ACRF Biomolecular Resource Facility, ANU). The plasmids were linearized with SalI (ThermoFisher Scientific) and 5'-capped complementary RNA (cRNA) was synthesized using the mMessage mMachine T7 transcription kit (Ambion), and then purified with the MEGAclear kit (Ambion).

### Harvesting and preparation of *Xenopus laevis* oocytes

Ethical approval of the work performed with the *Xenopus laevis* frogs (purchased from NASCO) was obtained from the Australian National University Animal Experimentation Ethics Committee (Animal Ethics Protocol Number A2016/12) in accordance with the Australian Code of Practice for the Care and Use of Animals for Scientific Purposes. Pieces of ovary were surgically removed from adult female frogs and treated with collagenase A and D (Roche) to yield single, de-folliculated oocytes.[74] Stage V-VI oocytes were microinjected with cRNA (20 ng per oocyte) encoding one of the PfCRT isoforms under study and were stored at 16–18˚C in $OR^{2+}$ buffer (82.5 mM NaCl, 2.5 mM KCl, 1 mM $MgCl_2$, 1 mM $Na_2HPO_4$, 5 mM HEPES, 1 mM $CaCl_2$; pH 7.8) supplemented with 50 μg/mL penicillin/streptomycin.

### Uptake of radiolabeled CQ by *Xenopus* oocytes

The transport of [$^3$H]CQ (American Radiolabeled Chemicals; 20 Ci/mmol) into oocytes was measured 3–5 days post-cRNA injection. The uptake assays were conducted over 1.5 h at 27.5˚C as previously described [24, 75]. Briefly, 10 oocytes expressing a given PfCRT isoform

were incubated with ND96 buffer (96 mM NaCl, 2 mM KCl, 1 mM MgCl$_2$, 1.8 mM CaCl$_2$, 10 mM MES, and 10 mM Tris-base; pH 5.5) containing 0.25 μM [$^3$H]CQ and 15 μM unlabeled CQ. The assay was terminated by removing the reaction buffer and washing the oocytes with ice-cold ND96 buffer. The oocytes were lysed in 10% SDS and combined with MicroScint-40 microscintillant (PerkinElmer). The radioactivity in each oocyte was measured with a MicroBeta$^2$ microplate liquid scintillation analyzer (PerkinElmer). In all cases, at least three independent experiments were performed (on different days and using oocytes from different frogs), and within each experiment measurements were made from 10 oocytes per treatment.

Note that non-expressing oocytes and those expressing wild-type PfCRT (PfCRT$^{3D7}$) take up [$^3$H]CQ to similar (low) levels via simple diffusion of the neutral species of the drug; this represents the 'background' level of [$^3$H]CQ accumulation in oocytes [33].

## Oocyte membrane preparation and western blot analysis

Semi-quantitative measurements of PfCRT abundance were performed by western blot analysis using a protocol described in detail elsewhere [24]. These assays ascertain whether expression of the different isoforms resulted in similar levels of PfCRT protein in the membranes of the oocytes. Protein samples prepared from oocyte membranes were separated on a 4–12% Bis-Tris SDS-polyacrylamide gel (Life Technologies) and transferred to a Protran 0.45 μM nitrocellulose membrane (Amersham, GE Healthcare Life Sciences). Total protein staining (Ponceau) was used to evaluate sample loading and efficiency of transfer as outlined previously [24]. The membranes were probed with a rabbit anti-PfCRT antibody (concentration of 1:4,000; GenScript) followed by horseradish peroxidase-conjugated goat anti-rabbit antibody (1:8,000; Life Technologies, cat. no. 656120). The specificity of the anti-PfCRT antibody has been validated elsewhere [24]. The PfCRT protein bands were detected by chemiluminescence (Pierce), quantified using ImageJ,[76] and expressed as a percentage of the band intensity measured for the PfCRT$^{Dd2}$ protein. In all cases, at least three independent experiments were performed (using oocytes from different frogs), and in each experiment the measurements were averaged from two independent replicates.

## Immunofluorescence of oocytes expressing PfCRT

Immunofluorescence analyses were performed on oocytes three days post-cRNA injection using a method adapted from Weise et al.[77] and described in detail elsewhere [75]. Briefly, the oocytes were fixed and permeabilized before incubation with the rabbit anti-PfCRT antibody (1:100; GenScript). The oocytes were then incubated with the Alexa Fluor 488 donkey anti-rabbit antibody (1:500; Molecular Probes, cat. no. A-21206) and embedded in an acrylic resin using the Technovit 7100 plastic embedding system (Kulzer). A microtome was used to obtain ~4 μm slices of the oocytes, which were mounted on microscope slides. Images of the slices were obtained with a Leica Sp5 inverted confocal laser microscope (Leica Microsystems) using the 63x objective. Excitation was achieved with a 488 nm argon laser and the emissions were captured using a 500–550 nm filter. The images were acquired using the Leica Application Suite Advanced Fluorescence software (Leica Microsystems). At least two independent experiments were performed (on oocytes from different frogs) for each oocyte type, within which slices were examined from at least three oocytes.

## Statistical analysis of the *Xenopus* oocyte data

Statistical comparisons were made using one-way ANOVAs in conjunction with Tukey's multiple comparisons test or with the Student's *t*-test. All errors cited in the text and shown in the figures represent the SEM. Significance was defined as $P < 0.05$.

## Supporting information

**S1 Fig. The transport of CQ via mutant variants of PfCRT *in situ* and in the *Xenopus* oocyte expression system.** The orientation of PfCRT in both the parasite's digestive vacuole (DV) membrane and in the oocyte plasma membrane is such that its N- and C-termini are located in the cytosol. Chloroquine (CQ) is a weak-base drug which, in its neutral form, can diffuse across the membranes of the parasitized erythrocyte and into the DV. Within the acidic environment of this compartment, CQ becomes protonated ($CQH^+$ and $CQH_2^{2+}$) and thereby accumulates via weak-base trapping. Protonated CQ is then effluxed from the DV, via mutant variants of PfCRT, into the parasite's cytosol. In the *Xenopus* oocyte system, tritiated CQ is added to the acidic extracellular solution and the protonated drug is then transported into the oocyte's cytosol via mutant variants of PfCRT. Diffusion of uncharged CQ into the oocyte also occurs, albeit to low levels. Note that in both scenarios, the direction of PfCRT-mediated CQ transport is from the DV lumen/extracellular solution into the cell cytosol. That is, CQ is translocated to the cytosolic compartment of the cell, which is also where the N- and C-termini of the transporter are located.
(TIF)

**S2 Fig. Combinatorial variants of PfCRT localize to the surface of *Xenopus* oocytes.** (Similar results for a second frog a shown in S3 Fig). Immunofluorescence microscopy was used to localize those PfCRT variants that exhibited little or no CQ transport activity when expressed in the oocyte system. In each case, the expression of the PfCRT variant resulted in a fluorescent band external to the pigment layer, indicating that the protein was expressed in the oocyte plasma membrane. The band was not present in non-expressing oocytes. Panels A and B show the images from two independent experiments that were performed using oocytes from two different frogs, and within which images were obtained from a minimum of three oocytes per PfCRT variant. Refer to Richards et al. [75] for the localization of IEKSESII (i.e. the '106/1' isoform of PfCRT).
(TIF)

**S3 Fig. Combinatorial variants of PfCRT localize to the surface of *Xenopus* oocytes.** (Similar results for a second frog a shown in S2 Fig). Immunofluorescence microscopy was used to localize those PfCRT variants that exhibited little or no CQ transport activity when expressed in the oocyte system. In each case, the expression of the PfCRT variant resulted in a fluorescent band external to the pigment layer, indicating that the protein was expressed in the oocyte plasma membrane. The band was not present in non-expressing oocytes. Panels A and B show the images from two independent experiments that were performed using oocytes from two different frogs, and within which images were obtained from a minimum of three oocytes per PfCRT variant. Refer to Richards et al. [75] for the localization of IEKSESII (i.e. the '106/1' isoform of PfCRT).
(TIF)

**S4 Fig. Distribution of % CQ transport activity.** Sub panels show % CQ transport for the 76 genotypes with measured phenotypes (top panel), the model predictions for those 76 measured phenotypes (middle panel) and the 180 genotypes with unmeasured phenotypes (bottom panel). % CQ transport was measured relative to PfCRT$^{Dd2}$.
(TIF)

**S5 Fig. Predictive epistatic coefficients cannot be resolved from experimental genotype-phenotype maps.** Each sub-panel shows $R^2_{train}$ (black) and $R^2_{test}$ (red) for the map indicated above the graph (see S2 Table) as epistatic orders are added to the model. The x-axis is the number of parameters used in the fit. Points are, from left to right: additive, pairwise, and

high-order epistasis. Points and lines indicate the mean of 1,000 pseudoreplicate samples in which we trained a model on 80% of the genotypes and predicted the remaining 20%. Error bars are standard deviation of pseudoreplicate results. The dashed lines indicate the fraction of the variation in the map explained by the additive model.
(TIF)

**S1 Table. Primer sequences used to introduce mutations into the PfCRT coding sequence via site-directed mutagenesis.**
(DOCX)

**S2 Table. Published genotype-phenotype maps used to test method.**
(DOCX)

**S1 File. Measured and predicted CQ transport activity for the complete genotype-phenotype map.**
(XLSX)

**S2 File. Calculated probabilities of all forward trajectories through the genotype-phenotype map.**
(XLSX)

**S3 File. Fit parameters for predictions on maps I-XII.**
(XLSX)

## Acknowledgments

We thank members of the Harms and Martin labs for helpful discussions and comments.

## Author Contributions

**Conceptualization:** Zachary R. Sailer, Rowena E. Martin, Michael J. Harms.

**Data curation:** Sarah H. Shafik, Robert L. Summers, Alex Joule, Rowena E. Martin.

**Funding acquisition:** Rowena E. Martin.

**Investigation:** Zachary R. Sailer, Sarah H. Shafik, Robert L. Summers, Alex Joule, Alice Patterson-Robert, Michael J. Harms.

**Methodology:** Michael J. Harms.

**Resources:** Zachary R. Sailer, Rowena E. Martin.

**Software:** Zachary R. Sailer, Michael J. Harms.

**Supervision:** Rowena E. Martin, Michael J. Harms.

**Validation:** Sarah H. Shafik, Robert L. Summers, Alex Joule, Alice Patterson-Robert.

**Visualization:** Zachary R. Sailer, Sarah H. Shafik, Rowena E. Martin, Michael J. Harms.

**Writing – original draft:** Zachary R. Sailer, Michael J. Harms.

**Writing – review & editing:** Zachary R. Sailer, Sarah H. Shafik, Robert L. Summers, Alex Joule, Rowena E. Martin, Michael J. Harms.

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
