## [Decision Letter · Decision Letter 0]

30 Jan 2020

Dear Prof Harms,

Thank you very much for submitting your manuscript "Inferring a complete genotype-phenotype map from a small number of measured phenotypes" for consideration at PLOS Computational Biology.

As with all papers reviewed by the journal, your manuscript was reviewed by members of the editorial board and by several independent reviewers. In light of the reviews (below this email), we would like to invite the resubmission of a significantly-revised version that takes into account the reviewers' comments.

We cannot make any decision about publication until we have seen the revised manuscript and your response to the reviewers' comments. Your revised manuscript is also likely to be sent to reviewers for further evaluation.

Sincerely,

Alexandre V. Morozov, Ph.D.

Associate Editor

PLOS Computational Biology

Feilim Mac Gabhann

Editor-in-Chief

PLOS Computational Biology

Reviewer's Responses to Questions

**Comments to the Authors:**

Reviewer #1: Review is attached as a plain text file.

Reviewer #2: Sailer et al. develop and analyze a method called GPSEER. This method

incorporates the additive effects of mutations, a binary classifier,

and a nonlinear scale, and is designed to predict missing phenotypes

from an incomplete combinatorial genotype-phenotype map. One of the

most impressive features of the method is its ability to characterize

the uncertainty of its predictions, which facilitates the

characterization of uncertainty in evolutionary forecasts. Sailer et

al. apply their method to a previously published combinatorial

genotype-phenotype map for the phenotype of chloroquine (CQ) transport

activity, which is important for the evolution of drug resistance in

the malaria parasite. They validate their model by using it to predict

the CQ transport activities of 24 new genotypes, which were measured

for this study. They then study the complete, predicted map to make

evolutionary inferences about the evolution of chloroquine (CQ)

transport activity in nature. The paper is exceptionally well written,

clear, interesting, and of direct relevance to the readership of PLoS

Computational Biology. The methods are also made freely available as a

command line tool and API. I commend the authors for their work.

My main concern with this paper is that it is about a new method, but

this new method is only tested on a single genotype-phenotype map. I

understand that this particular map is an important one to study and

that in applying their method to this map, the authors have gained

important insight into the evolution of chloroquine (CQ) transport

activity. This is all good. But if this paper is about a new method,

what have we learned about this method's predictive power outside of

this particular genotype-phenotype map? How does it perform on other

genotype-phenotype maps? Does its performance depend upon

topographical properties of the map, such as its overall ruggedness?

The authors state that their method "should be applicable to a large

number of genotype-phenotype maps", but whether or not it is remains

to be seen.

I have detailed this major concern below, as well as some other minor

concerns. Having been on the other side of a review like this before,

I completely understand if the authors disagree with and are

frustrated by my assessment. I therefore leave it up to them to decide

if the incorporation of more genotype-phenotype maps is necessary to

fairly assess the performance of their method, but if they decide

it is not, then I kindly ask for a justification.

Major:

My one major criticism of this otherwise excellent work is that the

authors only study one combinatorially complete genotype-phenotype

map, when many others are publicly available.

For example,

Bank et al. (2016) "On the (un)predictability of a large intragenic

fitness landscape" PNAS

In some cases, complete landscapes are available, which could be

subsampled to create combinatorially-complete landscapes in

replicate. For example,

Wu et al. (2016) "Adaptation in protein fitness landscapes is

facilitated by indirect paths" eLife

Alternatively, or additionally, the authors could apply their method

to simulated combinatorially complete genotype-phenotype maps.

By applying their method to these or other combinatorially-complete

landscapes (those listed above are just a few of many examples) the

authors could better assess its performance.

Minor:

- An important reference is missing:

du Plessis, Leventhal, & Bonhoeffer (2016) "How good are statistical

models at approximating complex fitness landscapes" Mol Biol Evol 33,

2454-2468.

I think this could be discussed in the final paragraph before the

discussion section "Future improvements".

Also, depending upon the authors' and PLoS' policies regarding the

citation of pre-prints, this highly relevant paper could be discussed:

Zhou & McCandlish (2019) "Minimum epistasis interpolation for

sequence-function relationships" bioRxiv

Trivia

(btw, it is very annoying to not have page numbers or line numbers in

this manuscript)

In the caption of Fig. 2, the yellow line for the spline fit in B is

not described in the caption.

In the main text, the authors mix up the red arrows and blue arrows

when describing Fig. 2

In the references, some titles have all words with their first letter

capitalized, others just the first word. Please fix.

Reviewer #3: In their manuscript entitled “Inferring a complete genotype-phenotype map from a small number of measured phenotypes” the authors present an approach to predict a complete genotype-phenotype map from an observed subset. The genotype space consists of all 256 mutational combinations of eight particular amino-acid changes of the malaria parasite’s PfCRT transporter that distinguish the wild-type 3D7 variant from a Dd2 isoform with increased chloroquine (CQ) transport activity, which in turn leads to greater CQ resistance of the parasite.

This is interesting and timely work, and the manuscript is well-written and mostly clear.

The CQ transport phenotype has been measured for 52 of the 256 genotypes in previous work. The authors use these results to train predictive models of increasing complexity until they reach an R^2 of 0.97 with a 70-parameter model. There is always a danger of overfitting with so many parameters (particularly when the number of observations is relatively small) and the authors duly report that an additional 24 observations were not predicted well by this complex model. A rather simpler model performed better, which means that the large R^2 for the 70-parameter model was indeed due to overfitting (as the authors also conclude).

The authors then try to show that the choice of these particular 52+24 observations does not bias the predictions by constructing random samples of 52 from these 76 and recalculating their model.

This is not entirely convincing, since the 24 were not randomly chosen from the remaining 204, but were strongly influenced by the model predictions based on the 52. It would have perhaps been advisable to select an additional random sample. Moreover the random samples that are performed show two slightly worrying features, namely that the nonlinear part of the model no longer improves anything for the random samples (indeed decreases the performance very slightly), and also that the R^2 now is somewhere around 0.70 (red curve in Fig. 4B). The fact that the nonlinear part does not add much is also not too surprising if one compares Fig. 2G and 2H, which look almost identical despite going from R^2 = 0.78 to R^2 = 0.90. The main difference between the two is the outlier around position (-30,5), which may well affect the R^2 quite strongly. But the sensitivity of R^2 and indeed the model training process to such an outlier decreases the confidence one can have in this approach.

The predictions for the 180 remaining genotypes appear to largely be quite low. This may of course be accurate, but given the problems with the model outlined above, and also given that the model may systematically underestimate large phenotype values (Fig. 2G & 2H) I worry that most of the 180 are simply predicted to have low phenotype values because they are far away from the training set. Most of the regions of the genotype map with high transport activity in Fig. 5 can already be anticipated in Fig. 1.

Information that (as far as I can see) is missing, is:

- where are the 24 observed phenotypes in Fig. 5? In general I find it somewhat misleading that the predicted and observed values are not distinguished. We should clearly be able to see the 52+24+108.

- what governs proximity in the representations used in Figs. 1 and 5? Is it in any way related to sequence distance (presumably, looking at the lines connecting the nodes, which seem to be minimised in length?) If so then this contributes to my worry, since we simply see the red regions in Fig. 1 grow a bit, which means that only genotypes close to observed phenotypes with high phenotype values are predicted to have high phenotype values as well.

- what is the distribution of predicted values for the 180 genotypes versus the distribution of the 52+24?

In conclusion my worry is that the model is overly reliant on the 52 original observations (the selection criteria of which are also not discussed much). Based on the fact that the additive model with classifier is basically the best model here, and that the 24 additional observations are chosen with that rationale of selecting “genotypes with different numbers of mutations that were predicted to exhibit substantially different levels of CQ transport activity” an additive model will likely only find substantially different levels of transport activity in the proximity of genotypes already exhibiting that activity. The 24 are therefore highly dependent on the 52, and the additive nature of the model means that we can’t have high confidence that there isn’t another pocket of high transport activity hiding among the 180. I think there’s a good chance there indeed isn’t, given that a third of genotype space has already been observed, but the authors clearly envisage applying this method to much larger spaces, where such hidden pockets would be much more likely.

It is interesting that the additive model is so successful - I do believe this is probably a useful model - and a pity in a way that the authors start with such a large set of observations and add further observations that aren’t truly independent. One could imagine an approach that starts with a random sample of observations that covers the genotype space in a fairly even way and then trains a model using this data and tests it on a further random sample before filling in the space completely.

One way to further validate their method in this particular case would be to measure the phenotype for a random sample (uniform in genotype space) of the 180, but I realise that this is probably a lot of additional work.

If the authors can provide convincing arguments that allay the above concerns, I’d be happy to recommend this article for publication.

**Have all data underlying the figures and results presented in the manuscript been provided?**

Reviewer #1: Yes

Reviewer #2: Yes

Reviewer #3: Yes

PLOS authors have the option to publish the peer review history of their article (what does this mean?). If published, this will include your full peer review and any attached files.

Reviewer #1: No

Reviewer #2: No

Reviewer #3: No
---

## [Decision Letter · Decision Letter 1]

1 Jul 2020

Dear Prof Harms,

Thank you very much for submitting your manuscript "Inferring a complete genotype-phenotype map from a small number of measured phenotypes" for consideration at PLOS Computational Biology. As with all papers reviewed by the journal, your manuscript was reviewed by members of the editorial board and by several independent reviewers. The reviewers appreciated the attention to an important topic. Based on the reviews, we are likely to accept this manuscript for publication, providing that you modify the manuscript according to the review recommendations.

Sincerely,

Alexandre V. Morozov, Ph.D.

Associate Editor

PLOS Computational Biology

Feilim Mac Gabhann

Editor-in-Chief

PLOS Computational Biology

[LINK]

Reviewer's Responses to Questions

**Comments to the Authors:**

Reviewer #3: The authors have revised - and I believe, improved - the paper substantially in light of the referees’ comments. I think this now merits publication pretty much as is. The only revision I would suggest is in the new section on the application of this model to other published genotype-phenotype maps, which I think is a great idea, but which is a little sparse and requires some details to be clarified:

- it is unclear to what extent the phenotypes in the 12 maps mentioned in Table S2 were measured. Were they all mapped fully? If not it would be nice to see the extent in the table.

- are the parameters for the application of the model to these other GP maps given somewhere?

- the references in line 4, p.19, “[4,10,28,57–60]” aren’t the same set of references as mentioned in Table S2, which are [4,10,19,59,60,83,84]. Is this correct?

- for the 13th, much larger map, how were the training and test sets chosen with respect to the underlying (skewed) distribution of the 59,394 oligonucleotides? Would different samplings that either (a) mirror the skew or (b) try to mitigate it by sampling in a way that ensures equal representation of the bases where possible change the results, and what would that mean for the method?

With these issues addressed I’m happy to recommend this paper for publication.

Reviewer #4: The review has been uploaded as an attachment.

**Have all data underlying the figures and results presented in the manuscript been provided?**

Reviewer #3: Yes

Reviewer #4: Yes

PLOS authors have the option to publish the peer review history of their article (what does this mean?). If published, this will include your full peer review and any attached files.

Reviewer #3: No

Reviewer #4: No
---

## [Editor Report · Decision Letter 2]

13 Aug 2020

Dear Prof Harms,

We are pleased to inform you that your manuscript 'Inferring a complete genotype-phenotype map from a small number of measured phenotypes' has been provisionally accepted for publication in PLOS Computational Biology.

Best regards,

Alexandre V. Morozov, Ph.D.

Associate Editor

PLOS Computational Biology

Feilim Mac Gabhann

Editor-in-Chief

PLOS Computational Biology

---

## [Editor Report · Acceptance letter]

23 Sep 2020

PCOMPBIOL-D-19-01733R2 

Inferring a complete genotype-phenotype map from a small number of measured phenotypes

Dear Dr Harms,

I am pleased to inform you that your manuscript has been formally accepted for publication in PLOS Computational Biology. Your manuscript is now with our production department and you will be notified of the publication date in due course.

With kind regards,

Matt Lyles
